# Inference of stress-strength reliability for two-parameter of exponentiated Gumbel distribution based on lower record values

**Ehsan Fayyazishishavan**[ORCID]**, Serpil Kılıç Depren***

Department of Statistics, Yildiz Technical University, Istanbul, Turkey

* serkilic@yildiz.edu.tr

## Abstract

The two-parameter of exponentiated Gumbel distribution is an important lifetime distribution in survival analysis. This paper investigates the estimation of the parameters of this distribution by using lower records values. The maximum likelihood estimator (MLE) procedure of the parameters is considered, and the Fisher information matrix of the unknown parameters is used to construct asymptotic confidence intervals. Bayes estimator of the parameters and the corresponding credible intervals are obtained by using the Gibbs sampling technique. Two real data set is provided to illustrate the proposed methods.

**Data Availability Statement:** All relevant data are within the manuscript and its Supporting Information files.

## 1 Introduction

In engineering applications, a system may be subjected to several stresses such as extreme temperature and pressure. The survival and performance of such systems strongly depend on their resistance strength. The models which try to measure this resistance are called stress-strength models, and in the simplest terms, it can be described as an evaluation of the experienced random "stress" ($Y$) and the available "strength" ($X$) which overcomes the stress. This simple explanation induces the definition of the reliability of a system as the probability that the system is strong enough to overcome the stress that it is subjected to. Therefore, the reliability parameter could be defined as $R = P(X > Y)$.

The estimation of the reliability parameter has extensive literature. It has been studied under different assumptions over the distribution of $X$ and $Y$. [1] studied the ML estimation of $R$ under the assumption that the stress and strength variables follow a bivariate exponential distribution. By considering a multivariate normal distribution, the MLE has been studied by [2]. The estimation of $R$, when the distribution is Weibull, were considered by [3]. See [4] and references therein for more details, works on the estimation of $R$ and its applications. In some recent works [5], estimated $R$ under the assumption that the stress and strength variables are independent and follow a generalized exponential distribution. [6] considered the estimation of $R$, when $X$ and $Y$ are independent, and both follow a three parameter Weibull distributions. Some other applications of the stress and strength models in the framework of transportation problems, which were estimated by ML methods, include [7–9]. Other engineer applications of these methods, which were applied in the Bayesian framework, could be found at [10–12].

**Funding:** The author(s) received no specific funding for this work.

**Competing interests:** The authors have declared that no competing interests exist.

The only difference in the above-mentioned works was the different distributions which the authors have been chosen for the random quantities. In some situations, one could not obtain the complete data and have to consider certain sampling schemes in order to get incomplete data for $X$ and $Y$. [13–15] have been studied the problem of making inference on $R$ based on progressively Type II censored data. [16], based on the record data, by considering one parameter generalized exponential distribution, has been studied ML and Bayesian estimation of $R$.

Another type of incomplete data is record values which usually appear in many real-life applications. Record values arise in climatology, sports, business, medicine, industry, and life testing surveys, among others. These records are commemorating over the period of the time that have been studied. The history of records may show the advancement in science and technology. By considering the record values in various areas of humankind activities, we can evaluate the performance of the societies. In 1952, [17] introduced the distribution of record values into the statistical world. After six decades of his original work, hundreds of papers were devoted to various aspects of the record's theory. He provided a foundation for a mathematical theory of records. He defined the record values as consecutive extremes appearing in a sequence of independent identically distributed (*i.i.d.*) random variables. These smallest or largest occurred values are called "lower" or "upper" record values, respectively. Let $X_1, \ldots, X_n$ be a sequence of *i.i.d.* continuous random variables with a cumulative distribution function (CDF) $F(x)$ and its corresponding probability density function (PDF) $f(x)$. For every positive integer $k \geq 1$, the sequence of $k$th lower record times, $\{L(k), k \geq 1\}$, is defined as follows

$$L(1) = 1, \quad L(k + 1) = \min\{j | j > L(k), X_j < X_{L(k)}\}. \tag{1}$$

Then the $k$th lower record value will be denoted by $X_{L(k)}$ and the sequence $\{X_{L(k)}, L(k) \geq 1\}$ is called the lower record values. For the sake of simplicity, from now on, we shall refer to $X_{L(k)}$ as $X_k$. As mentioned before, the record values have many applications in industry and engineering. Consider an electronic system that is subject to some shocks like low or high voltage in which both are dangerous for its predefined performance. These shocks could be considered as realizations of *i.i.d.* variable, and then one can use the record values models to study them. We refer the readers to [18, 19] for more details on the record values and their applications.

The Gumbel distribution is a well-known and popular model due to its wide application in climatology, global warming problems, wind speed, and rainfall modeling, among others. The book of [20] has an extensive list of applications of the Gumbel distribution in various fields of science. [21] has generalized this distribution by exponentiating, in the form of $F(x;\alpha) = 1 - [1 - G(x)]^{\alpha}$, where $G(x)$ is the Gumbel density and $a > 0$. Note that exponentiating the standard probability distributions cloud solves the problem of lack of fits that arise when using these distributions for modeling complex data [22]. They showed the power and ability of this generalized distribution in modeling the climatology data by applying it on rainfall data from Orlando, Florida.

In this work, we use a slightly different way to define the exponentiated distributions, i.e., $F(x;\alpha) = [G(x)]^{\alpha}$, which are called the proportional reversed hazard rate models [23]. The random variable $X$ follows the two-parameter Exponentiated Gumbel distribution if it has the following CDF

$$F(x; \alpha, \lambda) = e^{-\alpha e^{-\lambda x}}, \quad -\infty < x < +\infty \tag{2}$$

where $\alpha > 0$ and $\lambda > 0$. The PDF corresponding to the CDF (2) is

$$f(x; \alpha, \lambda) = \alpha \lambda e^{-\lambda x} e^{-\alpha e^{-\lambda x}}, \quad -\infty < x < +\infty. \tag{3}$$

Here $\alpha$ and $\lambda$ are the shape and scale parameters, respectively. We will denote this distribution by $EG(\alpha, \lambda)$.

## 2 Maximum likelihood estimation

In this section, we consider the maximum likelihood estimation of $R = P(X > Y)$ when $X \sim EG(\alpha, \lambda)$ and $Y \sim EG(\beta, \lambda)$, and $X$ and $Y$ are independently distributed. Formal integration shows that

$$R = P(X > Y) = \int_{-\infty}^{\infty} \int_{-\infty}^{x} \alpha \beta \lambda^2 e^{-\lambda x} e^{-\alpha e^{-\lambda x}} e^{-\lambda y} e^{-\beta e^{-\lambda y}} dy\, dx = \frac{\alpha}{\alpha + \beta}. \tag{4}$$

Let $X_1, X_2, \ldots, X_n$ and $Y_1, Y_2, \ldots, Y_m$ be two independent sets of the lower records from $EG(\alpha, \lambda)$ and $EG(\beta, \sigma)$, respectively. Therefore, the likelihood function of parameters becomes (see [19])

$$L(\alpha, \beta, \lambda) = \left[ f(x_n; \alpha, \lambda) \prod_{k=1}^{n-1} \frac{f(x_k; \alpha, \lambda)}{F(x_k; \alpha, \lambda)} \right] \left[ f(y_m; \beta, \lambda) \prod_{j=1}^{m-1} \frac{f(y_j; \beta, \lambda)}{F(y_j; \beta, \lambda)} \right]$$

From (2) and (3) the likelihood function is obtained as

$$L(\alpha, \beta, \lambda) = \alpha^n \beta^m \lambda^{(m+n)} \exp \left\{ -\lambda \left( \sum_{k=1}^{n} x_k + \sum_{j=1}^{m} y_j \right) - \alpha e^{-\lambda x_n} - \beta e^{-\lambda y_m} \right\}.$$

The log likelihood function is given by

$$\ell(\alpha, \beta, \lambda) = n\ln\alpha + m\ln\beta + (m+n)\ln\lambda - \lambda \left( \sum_{k=1}^{n} x_k + \sum_{j=1}^{m} y_j \right) - \alpha e^{-\lambda x_n} - \beta e^{-\lambda y_m}.$$

Then, the likelihood equations will be

$$\frac{\partial \ell}{\partial \alpha} = \frac{n}{\alpha} - e^{-\lambda x_n} = 0 \tag{5}$$

$$\frac{\partial \ell}{\partial \beta} = \frac{n}{\beta} - e^{-\lambda y_m} = 0 \tag{6}$$

$$\frac{\partial \ell}{\partial \lambda} = \frac{n+m}{\lambda} - \sum_{k=1}^{n} x_k - \sum_{j=1}^{m} y_j + \alpha x_n e^{-\lambda x_n} + \beta y_m e^{-\lambda y_m} = 0 \tag{7}$$

From above equations, we get

$$\hat{\lambda} = \frac{n+m}{n(\bar{x} - x_n) + m(\bar{y} - y_m)}, \quad \hat{\alpha} = n e^{\hat{\lambda} x_n}, \quad \hat{\beta} = m e^{\hat{\lambda} y_m}. \tag{8}$$

Note that $\hat{\lambda}$ is the harmonic mean of $\hat{\lambda}_1 = 1/(\bar{x} - x_n)$ and $\hat{\lambda}_2 = 1/(\bar{y} - y_m)$, which are the MLEs of independent samples of sizes $n$ and $m$, respectively.

Therefore, by applying the invariant property of ML estimators, the ML estimation of $R$ will be

$$\hat{R} = \frac{\hat{\alpha}}{\hat{\alpha} + \hat{\beta}}. \tag{9}$$

In this section, we obtain the Fisher information matrix of the unknown parameters of EG distribution, which can be used to construct asymptotic confidence intervals.

[19] showed that the PDF of the $s$th lower record, $X_s$, is given by

$$f_{X_s}(x) = \frac{f(x)[-\ln F(x)]^{s-1}}{\Gamma(s)} = \frac{1}{\Gamma(s)} \alpha^s \lambda e^{-s\lambda x - \alpha e^{-\lambda x}}, \quad x \in \mathbb{R}.$$

Therefore, the Fisher information matrix of $\boldsymbol{\theta} = (\alpha, \beta, \lambda)$ will be

$$I(\boldsymbol{\theta}) = - \begin{bmatrix} E\left(\frac{\partial^2 \ell}{\partial \alpha^2}\right) & E\left(\frac{\partial^2 \ell}{\partial \alpha \partial \beta}\right) & E\left(\frac{\partial^2 \ell}{\partial \alpha \partial \lambda}\right) \\[2ex] E\left(\frac{\partial^2 \ell}{\partial \beta \partial \alpha}\right) & E\left(\frac{\partial^2 \ell}{\partial \beta^2}\right) & E\left(\frac{\partial^2 \ell}{\partial \beta \partial \lambda}\right) \\[2ex] E\left(\frac{\partial^2 \ell}{\partial \lambda \partial \alpha}\right) & E\left(\frac{\partial^2 \ell}{\partial \lambda \partial \beta}\right) & E\left(\frac{\partial^2 \ell}{\partial \lambda^2}\right) \end{bmatrix}$$

where

$$I_{11} = \frac{n}{\alpha^2}, \quad I_{12} = I_{21} = 0, \quad I_{13} = I_{31} = \frac{n}{\alpha\lambda}(\psi(n+1) - \ln\alpha), \quad I_{22} = \frac{m}{\beta^2},$$

$$I_{23} = I_{32} = \frac{m}{\beta\lambda}(\psi(m+1) - \ln\beta),$$

$$I_{33} = \frac{m+n}{\lambda^2} + \frac{n}{\lambda^2}\left[\psi'(n+1) + (\psi(n+1) - \ln\alpha)^2\right] + \frac{m}{\lambda^2}\left[\psi'(m+1) + (\psi(m+1) - \ln\beta)^2\right].$$

The asymptotic covariance matrix of the ML estimators could be achieved via inverting the Fisher information matrix as following

$$I^{-1}(\boldsymbol{\theta}) = U(\boldsymbol{\theta}) = \frac{1}{\det(I(\boldsymbol{\theta}))} \begin{bmatrix} I_{22}I_{33} - I_{23}^2 & I_{13}I_{32} & -I_{13}I_{22} \\[2ex] I_{23}I_{31} & I_{11}I_{33} - I_{13}^2 & -I_{11}I_{23} \\[2ex] -I_{22}I_{31} & -I_{11}I_{32} & I_{11}I_{22} \end{bmatrix},$$

where

$$\det(I(\boldsymbol{\theta})) = I_{11}I_{22}I_{33} - I_{11}I_{23}^2 - I_{22}I_{13}^2.$$

Now, by using the delta method, the asymptotic variance of $\hat{R}$ could be obtained as follows.

$$\mathbb{V}(\hat{R}) = C^T U C,$$

where $C^T = \left(\frac{\partial R}{\partial \alpha}, \frac{\partial R}{\partial \beta}, \frac{\partial R}{\partial \lambda}\right) = \frac{1}{(\alpha+\beta)^2}(\beta, -\alpha, 0)$.

Note that the $\mathbb{V}(\hat{R})$ is a function of unknown parameters, and it needs to be estimated. It can be done by plunging the ML estimators of the parameters. Therefore, the $(1-\gamma)\%$ asymptotic confidence intervals of $R$ will be in the form of $\hat{R} \pm z_{1-\gamma/2}\frac{\sqrt{\mathbb{V}(\hat{R})}}{\sqrt{n}}$.

## 3 Bayesian estimation

In this section, we attempt to find the Bayes estimator of the parameters. To do so, we consider that the parameters are apriori independent, and they follow gamma distributions, i.e., $\alpha \sim Gamma(a_1, b_1)$, $\beta \sim Gamma(a_2, b_2)$, and $\lambda \sim Gamma(a_3, b_3)$. Therefore, the full posterior distribution of the parameters will be

$$\pi(\alpha, \beta, \lambda | data) \propto \alpha^n \beta^m \lambda^{(m+n)} \exp\{-\lambda(\textstyle\sum_{k=1}^{n} x_k + \sum_{j=1}^{m} y_j) - \alpha e^{-\lambda x_n} - \beta e^{-\lambda y_m}\} \tag{10}$$

$$\times \alpha^{a_1-1} e^{-b_1 \alpha} \beta^{a_2-1} e^{-b_2 \beta} \lambda^{a_3-1} e^{-b_3 \lambda} \tag{11}$$

The above posterior does not admit a closed-form and cannot be used directly in the estimation procedure. Then to simulate a random sample from such distributions and perform an approximated inference, the Gibbs sampler could be used. The full conditional distributions of the parameters are as follows:

$$\alpha | \beta, \lambda, data) \sim Gamma(n + a_1, b_1 + e^{-\lambda x_n}),$$

$$\beta | \alpha, \lambda, data) \sim Gamma(m + a_2, b_2 + e^{-\lambda y_m}),$$

$$\pi(\lambda | \alpha, \beta, data) \propto \lambda^{m+n+a_3-1} \exp\{-\lambda(b_3 + n\bar{x} + m\bar{y}) - \alpha e^{-\lambda x_n} - \beta e^{-\lambda y_m}\}.$$

As $\pi(\lambda | \alpha, \beta, data)$ does not have a closed and standard form, one could not produce a sample from this density using direct methods. The Metropolis-Hastings algorithm is a method that can be used to produce a sample from such distributions. As shown in Fig 1, the normal distribution could be a good candidate for the proposal distribution of the Metropolis-Hastings algorithm. Therefore, the algorithm of Gibbs sampling is as follows.

- Step 1: Start with an initial value $\lambda^{(0)}$.

- Step 2: Set $t = 1$.

- Step 3: Generate $\alpha^{(t)}$ from $Gamma(n + a_1, b_1 + e^{-\lambda^{(t-1)} x_n})$.

- Step 4: Generate $\beta^{(t)}$ from $Gamma(m + a_2, b_2 + e^{-\lambda^{(t-1)} y_m})$,.

- Step 5: Use the Metropolis-Hastings algorithm to generate $\lambda^{(t)}$ from $\pi(\lambda | \alpha^{(t-1)}, \beta^{(t-1)}, data)$ by using the $N(\lambda^{(t-1)}, \sigma_0^2)$ as a proposal distribution.

  - Step 5.1: Generate candidate points $\lambda^*$ from $N(\lambda^{(t-1)}, \sigma_0^2)$ and $u$ from $\mathcal{U}(0, 1)$.

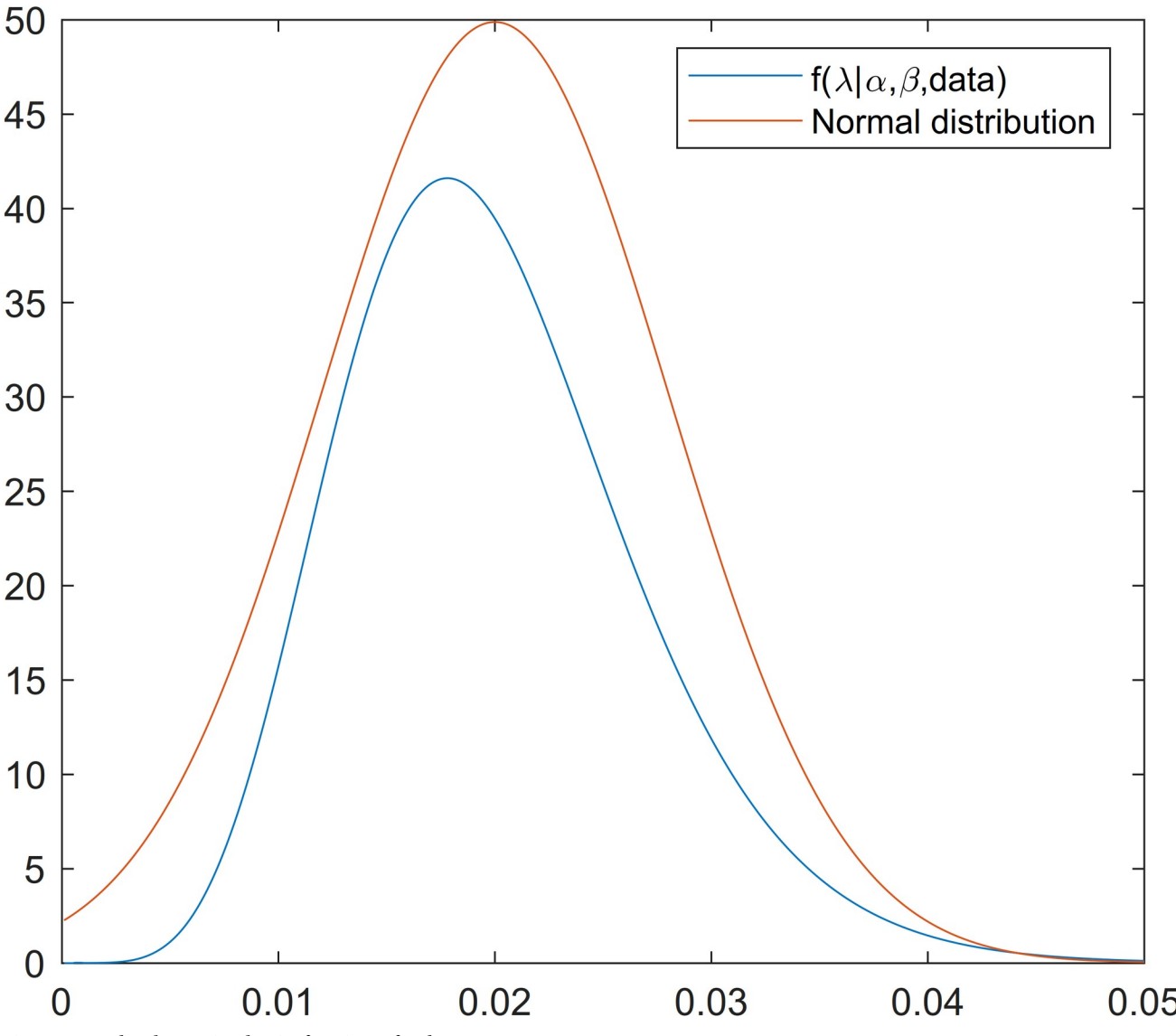

**Fig 1. Proposal and posterior density functions of scale parameter.**

- Step 5.2: Set $\lambda^{(t)} = \lambda^*$ if $u \leq \rho(\lambda^{(t-1)}, \lambda^*)$ and $\lambda^{(t)} = \lambda^{(t-1)}$ otherwise, when the acceptance probability is given by $\rho(\lambda^{(t-1)}, \lambda^*) = \min\{1, A\}$, and the acceptance rate is given by

$$A = \frac{\pi(\lambda^* | \alpha^{(t)}, \beta^{(t)}, data)}{\pi(\lambda^{(t-1)} | \alpha^{(t)}, \beta^{(t)}, data)} \cdot \frac{N(\lambda^{(t-1)} | \lambda^*, \sigma_0^2)}{N(\lambda^* | \lambda^{(t-1)}, \sigma_0^2)}$$

$$= \frac{\pi(\lambda^* | \alpha^{(t)}, \beta^{(t)}, data)}{\pi(\lambda^{(t-1)} | \alpha^{(t)}, \beta^{(t)}, data)}.$$

- Step 6: Set $t = t+1$.

- Step 7: Repeat steps 3–6, $T$ times.

Once we get a sample from the posteriors, the approximate posterior mean of $R$, and its variance could be computed as following

$$\hat{E}(R|data) = \frac{1}{T-K} \sum_{t=K+1}^{T} R^{(t)} = \frac{1}{T-K} \sum_{t=K+1}^{T} \frac{\alpha^{(t)}}{\alpha^{(t)} + \beta^{(t)}}$$

and

$$\hat{V}ar(R|data) = \frac{1}{T-K} \sum_{t=K+1}^{T} \left(R^{(t)} - \hat{E}(R|data)\right)^2$$

where $K$ is the burn-in period of the chain, which helps to vanish the effect of the starting values of the generated Markov chain.

The approximate highest posterior density (HPD) credible interval of $R$ could be constructed using the method proposed in [24].

Let $R_{(K+1)} < R_{(K+2)} < \cdots < R_{(T)}$ be the ordered output of the chain, $R^{(t)}$. To construct a $100(1-\gamma)\%$ approximate HPD credible interval for $R$, we consider the following intervals,

$$\{(R_{(K+1)}, R_{[((1-\gamma)T)]}), (R_{(K+2)}, R_{[((1-\gamma)T+1)]}), \dots, (R_{[(K+\gamma T)]}, R_{(T)})\},$$

by choosing the interval with the shortest length, we obtain the HPD credible intervals.

## 4 Inference on $R$ when $\lambda$ is known

As we show in section 2, the ML estimation of $\lambda$ does not depend on the value of other parameters; therefore, by plunging the MLE of $\lambda$, one can assume that the model contains only two parameters. This assumption makes the procedure of inference easier and straightforward. In other words, we can assume that $\lambda$ is known, and without loss of generality, we set $\lambda = 1$.

### 4.1 MLE of $R$

As mentioned in section 2, the ML estimator of $R$ is

$$\hat{R} = \frac{\hat{\alpha}}{\hat{\alpha} + \hat{\beta}} = \frac{n\,e^{X_n}}{n\,e^{X_n} + m\,e^{Y_m}}. \tag{12}$$

By straight computation, one can see that

$$2\alpha\,e^{-X_n} \sim \chi_{2n}^2 \quad \text{and} \quad 2\beta\,e^{-Y_m} \sim \chi_{2m}^2 \tag{13}$$

By considering (13), and the fact that $X_n$ and $Y_m$ are independent, one can show that

$$\frac{R}{1-R} \cdot \frac{1-\hat{R}}{\hat{R}} \sim F_{2n,2m}.$$

Therefore, the $100(1-\gamma)\%$ confidence interval for $R$ could be obtained as

$$\left(\frac{1}{1 + \frac{m\,e^{Y_m}}{n\,e^{X_n}} F_{1-\frac{\gamma}{2},2m,2n}}, \frac{1}{1 + \frac{m\,e^{Y_m}}{n\,e^{X_n}} F_{\frac{\gamma}{2},2m,2n}},\right) \tag{14}$$

where $F_{\gamma,d_1,d_2}$ is the $100\gamma^{th}$ percentile of the Fisher distribution with $d_1$ and $d_2$ degrees of freedom.

### 4.2 Bayesian estimation

Since we assumed that the parameters are apriori independent with gamma density, the posterior density of $\alpha$ and $\beta$ are independent $Gamma(a_1 + n, b_1 + e^{-x_n})$ and $Gamma(a_2 + m, b_2 + e^{-y_m})$, respectively. Therefore, the posterior distribution of $R$ will be

$$\pi(R|data) = c.r^{a_1 + n - 1}(1 - r)^{a_2 + m - 1}(1 - zr)^{-(a_1 + a_2 + n + m)}, \quad 0 < r < 1, \tag{15}$$

where

$$c = \frac{\Gamma(a_1 + a_2 + n + m)}{\Gamma(a_1 + n)\Gamma(a_2 + m)} \cdot \left(\frac{b_1 + e^{-x_n}}{b_2 + e^{-y_m}}\right)^{a_1 + n} \quad \text{and} \quad z = 1 - \frac{b_1 + e^{-x_n}}{b_2 + e^{-y_m}}. \tag{16}$$

The Bayesian estimation is based on the obtained posterior distribution. According to the assumed loss function, various aspects of the posterior distribution, such as the mean, median, etc., can be used to estimate the parameters. See [25, 26] for more details. By assuming the quadratic loss function, the Bayesian estimation will be the posterior mean which could be computed by considering the following well-known equation

$$B(b, c - b)_2F_1(a, b; c; z) = \int_0^1 x^{b-1}(1 - x)^{c-b-1}(1 - zx)^{-a}dx \quad c > b > 0, \tag{17}$$

in which $B(b,c−b)$ and $_2F_1(a,b; c; z)$ are beta and hypergeometric functions, receptively. Therefore, the Bayesian estimation of $R$ is

$$\hat{R}_{Bayes} = E(R|data) \tag{18}$$

$$= \frac{\Gamma(a_1 + a_2 + n + m)}{\Gamma(a_1 + n)\Gamma(a_2 + m)} \cdot \left(\frac{b_1 + e^{-x_n}}{b_2 + e^{-y_m}}\right)^{a_1 + n} B(a_1 + n + 1, a_2 + m)$$

$$\times {}_2F_1\left(a_1 + a_2 + n + m, a_1 + n + 1; a_1 + a_2 + n + m + 1; 1 - \frac{b_1 + e^{-x_n}}{b_2 + e^{-y_m}}\right).$$

The variance of the Bayesian estimator could be achieved by using

$$E(R^2|data) = \frac{\Gamma(a_1 + a_2 + n + m)}{\Gamma(a_1 + n)\Gamma(a_2 + m)} \cdot \left(\frac{b_1 + e^{-x_n}}{b_2 + e^{-y_m}}\right)^{a_1 + n} B(a_1 + n + 2, a_2 + m)$$

$$\times {}_2F_1\left(a_1 + a_2 + n + m, a_1 + n + 2; a_1 + a_2 + n + m + 2; 1 - \frac{b_1 + e^{-x_n}}{b_2 + e^{-y_m}}\right).$$

To construct the HPD intervals, as the posterior is not tractable, we can generate a sample from the posterior by using an indirect sampling algorithm, such as the accept-reject method.

### 4.3 Real data analysis

In this section, we analyze a set of real strength data, which were taken from ([27], p. 574). These data are originally from [28], which represent the lifetimes of steel specimens tested at 14 different stress magnitudes. Here, we pick up the dataset corresponding to 35.0 and 35.5 stress levels as Dataset 1 (x) and Dataset 2 (y) in Table 1, respectively.

We fitted the EG distribution models for two datasets separately and estimated the scale and shape parameters. The Kolmogorov-Smirnov (K-S) goodness of fit test was applied on the datasets. The reported results in Table 2 confirm the well-fitting of the EG distribution to

**Table 1. The real datasets which were reported in Lawless (2011) [27].**

| Dataset 1 (x) | | | | | Dataset 2 (y) | | | | |
|---|---|---|---|---|---|---|---|---|---|
| 230 | 169 | 178 | 271 | 129 | 156 | 173 | 125 | 852 | 559 |
| 568 | 115 | 280 | 305 | 326 | 442 | 168 | 286 | 261 | 227 |
| 1101 | 285 | 734 | 177 | 493 | 285 | 253 | 166 | 133 | 309 |
| 218 | 342 | 431 | 143 | 381 | 247 | 112 | 202 | 365 | 702 |

Dataset 1 (x) and Dataset 2 (y) correspond to 35.0 and 35.5 stress level, respectively.

**Table 2. The Kolmogorov-Smirnov test output for real datasets.**

| Dataset | Scale | Shape | K-S | p-value |
|---|---|---|---|---|
| 1 | 0.0071 | 5.9717 | 0.1137 | 0.9326 |
| 2 | 0.0085 | 6.5564 | 0.1408 | 0.7726 |

model these data. Moreover, Figs 2 and 3 confirm the appropriate fit of the EG distribution by comparing the empirical and fitted distributions for both datasets.

Now we can use the lower records based on the datasets to drive the ML and Bayesian estimation of parameters. These records for Dataset 1 are 230,169,129,115 and based on Dataset 2 are 156, 125, 112. The corresponding results for ML methods are reported in Table 2. According to this table, it is clear that the scale parameters of the two data sets are almost the same. By assuming equality of the scale parameter, the MLE and the 95% confidence interval of $R$ based on lower records values become 0.5927 and (0.4117,0.7737), respectively. Also, by using Gibs

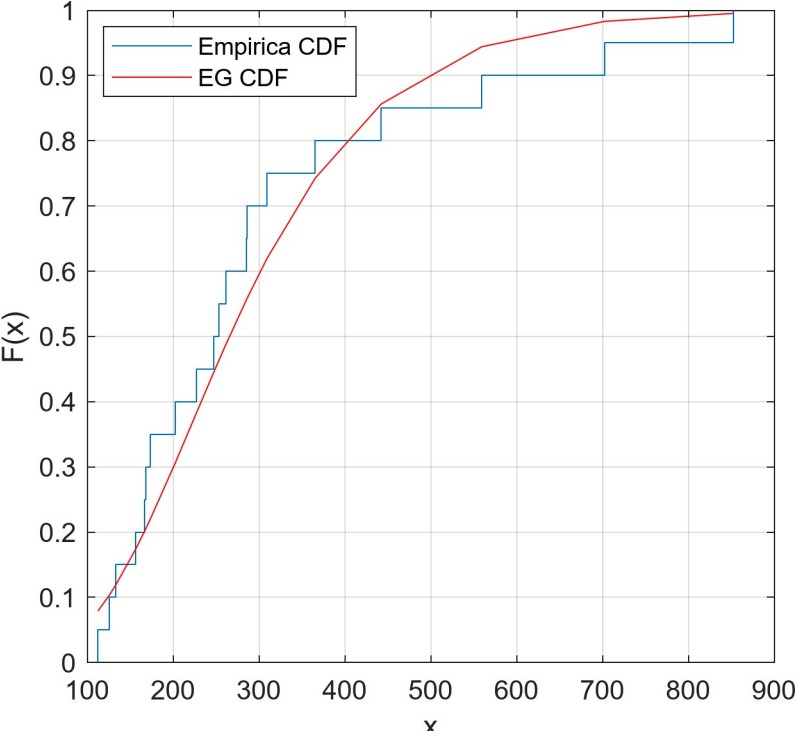

**Fig 2. Empirical and fitted CDFs for Dataset 1.**

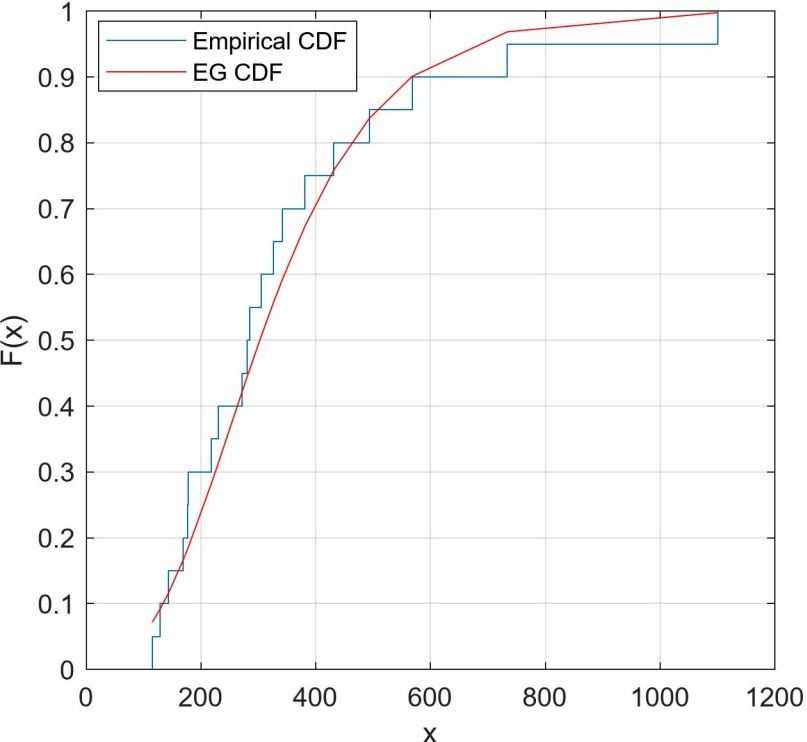

**Fig 3. Empirical and fitted CDFs for Dataset 2.**

sampling the Bayes estimate and credible interval of $R$ are 0.5893 and (0.4236,0.7641), respectively.

## 5 Conclusion

In this paper, we investigated the estimation of the parameters of the two-parameter of exponentiated Gumbel distribution by using lower records values. The maximum likelihood was used to estimate the parameters of the models, and the Fisher information matrix of the unknown parameters is used to construct asymptotic confidence intervals. Furthermore, the Bayes estimator of the parameters and the corresponding credible intervals were obtained by using the Gibbs sampling technique. The methods of estimating (ML and Bayes) were compared via two real data set and showed that the Bayesian estimations are slightly different from the ML ones.

## Supporting information

**S1 Dataset. Minimal dataset.**
(DOCX)

## Author Contributions

**Supervision:** Serpil Kılıç Depren.

**Writing – original draft:** Ehsan Fayyazishishavan.

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
