## [Decision Letter · Decision Letter 0]

3 Feb 2021

PONE-D-20-39915

Inference of Stress-Strength Reliability for Two-Parameter of Exponentiated Gumbel Distribution Based on Lower Record Values

PLOS ONE

Dear Dr. Kılıç Depren,

Thank you for submitting your manuscript to PLOS ONE. After careful consideration, we feel that it has merit but does not fully meet PLOS ONE’s publication criteria as it currently stands. Therefore, we invite you to submit a revised version of the manuscript that addresses the points raised during the review process.

We look forward to receiving your revised manuscript.

Kind regards,

Feng Chen

Academic Editor

PLOS ONE

Reviewers' comments:

Reviewer's Responses to Questions

**Comments to the Author**

1. Is the manuscript technically sound, and do the data support the conclusions?

Reviewer #1: Yes

Reviewer #2: Partly

2. Has the statistical analysis been performed appropriately and rigorously? 

Reviewer #1: Yes

Reviewer #2: Yes

3. Have the authors made all data underlying the findings in their manuscript fully available?

Reviewer #1: Yes

Reviewer #2: Yes

4. Is the manuscript presented in an intelligible fashion and written in standard English?

Reviewer #1: Yes

Reviewer #2: Yes

5. Review Comments to the Author

Reviewer #1: This paper proposes a Two-Parameter of exponentiated Gumbel distribution for modeling stress-strength reliability, based on lower record values. The research topic is interesting and worth of investigation. The proposed model is promising and its strengths have been demonstrated by two real-word datasets. A minor suggestion is that more works on Bayesian estimation and its engineer applications should be referred, such as:

A Bayesian spatial random parameters Tobit model for analyzing crash rates on roadway segments. Accident Analysis and Prevention, 2017, 100: 37-43.

Bayesian spatial-temporal model for the main and interaction effects of roadway and weather characteristics on freeway crash incidence. Accident Analysis and Prevention, 2019, 132, 105249.

Spatial joint analysis for zonal daytime and nighttime crash frequencies using a Bayesian bivariate conditional autoregressive model. Journal of Transportation Safety and Security, 2020, 12(4): 566-585.

The authors are also suggested to add a Conclusion section to draw some remarkable findings and present some directions for future research.

Reviewer #2: The topic of this paper is interesting. The methods sound. The results are meaningful and useful. There are some suggestions to improve this paper.

1. More reference of maximum likelihood is needed. For example, the following ones.

[1]  “Investigating the Differences of Single- and Multi-vehicle Accident Probability Using Mixed Logit Model", Journal of Advanced Transportation, 2018, UNSP 2702360.

[2] “Injury severities of truck drivers in single- and multi-vehicle accidents on rural highway”, Accident Analysis and Prevention, 2011, 43(5), 1677-1688.

[3] Analysis of hourly crash likelihood using unbalanced panel data mixed logit model and real-time driving environmental big data. 2018, JOURNAL OF SAFETY RESEARCH. 65: 153-159.

[2] A conclusion part is needed to summarize this paper.

6. PLOS authors have the option to publish the peer review history of their article (what does this mean?). If published, this will include your full peer review and any attached files.

Reviewer #1: No

Reviewer #2: No

---

## [Author Response · Author response to Decision Letter 0]

17 Feb 2021

RESPONSES TO THE REVIEWERS’ COMMENTS

We would like to express our gratitude to the Editor and Reviewers for their valuable comments and contributions to improve our study. We have replied to the comments point-by-point and revised the manuscript by considering the suggestions and recommendations.

Here are the answers to reviewers’ suggestions and revised parts are shown as track changes in the manuscript.

Reviewers' comments:

Comments to the Author

Comment 1: Is the manuscript technically sound, and do the data support the conclusions?

Reviewer #1: Yes

Reviewer #2: Partly

Comment 2: Has the statistical analysis been performed appropriately and rigorously?

Reviewer #1: Yes

Reviewer #2: Yes

Comment 3: Have the authors made all data underlying the findings in their manuscript fully available?

Reviewer #1: Yes

Reviewer #2: Yes

Comment 4: Is the manuscript presented in an intelligible fashion and written in standard English?

Reviewer #1: Yes

Reviewer #2: Yes

Comment 5: Review Comments to the Author

Reviewer #1: This paper proposes a Two-Parameter of exponentiated Gumbel distribution for modeling stress-strength reliability, based on lower record values. The research topic is interesting and worth of investigation. The proposed model is promising and its strengths have been demonstrated by two real-word datasets. A minor suggestion is that more works on Bayesian estimation and its engineer applications should be referred, such as:

A Bayesian spatial random parameters Tobit model for analyzing crash rates on roadway segments. Accident Analysis and Prevention, 2017, 100: 37-43.

Bayesian spatial-temporal model for the main and interaction effects of roadway and weather characteristics on freeway crash incidence. Accident Analysis and Prevention, 2019, 132, 105249.

Spatial joint analysis for zonal daytime and nighttime crash frequencies using a Bayesian bivariate conditional autoregressive model. Journal of Transportation Safety and Security, 2020, 12(4): 566-585.

The authors are also suggested to add a Conclusion section to draw some remarkable findings and present some directions for future research.

Answer: In line with the referee’s comment, reference list is revised. Also, Conclusion section has been added.

Reviewer #2: The topic of this paper is interesting. The methods sound. The results are meaningful and useful. There are some suggestions to improve this paper.

1. More reference of maximum likelihood is needed. For example, the following ones.

[1] “Investigating the Differences of Single- and Multi-vehicle Accident Probability Using Mixed Logit Model", Journal of Advanced Transportation, 2018, UNSP 2702360.

[2] “Injury severities of truck drivers in single- and multi-vehicle accidents on rural highway”, Accident Analysis and Prevention, 2011, 43(5), 1677-1688.

[3] Analysis of hourly crash likelihood using unbalanced panel data mixed logit model and real-time driving environmental big data. 2018, JOURNAL OF SAFETY RESEARCH. 65: 153-159.

[2] A conclusion part is needed to summarize this paper.

Answer: In line with the referee’s comment, reference list is revised. Also, Conclusion section has been added.

Comment 6: PLOS authors have the option to publish the peer review history of their article (what does this mean?). If published, this will include your full peer review and any attached files. 

Do you want your identity to be public for this peer review? For information about this choice, including consent withdrawal, please see our Privacy Policy.

Reviewer #1: No

Reviewer #2: No

Dear Editor,

Firstly, we want to thank you for your time and considerations. We have shown the revised parts as track changes in the file that is named as “Revised Manuscript with Track Changes”. We have modified the manuscript based on all required changes you mentioned, including:

1- We have referred to Bayesian works in our work, especially those references mentioned by reviewers. 

2- We have referred to more references of maximum likelihood works in our work, especially those you mentioned. 

3- We have added a Conclusion section to draw some remarkable findings and present some directions for future research.

We would like to present the updated version of our manuscript for evaluation. We are looking forward to getting your final decision on our work. 

Best regards,

---

## [Decision Letter · Decision Letter 1]

10 Mar 2021

Inference of Stress-Strength Reliability for Two-Parameter of Exponentiated Gumbel Distribution Based on Lower Record Values

PONE-D-20-39915R1

Dear Dr. Kılıç Depren,

We’re pleased to inform you that your manuscript has been judged scientifically suitable for publication and will be formally accepted for publication once it meets all outstanding technical requirements.

Kind regards,

Feng Chen

Academic Editor

PLOS ONE

Additional Editor Comments (optional):

Reviewers' comments:

Reviewer's Responses to Questions

**Comments to the Author**

1. If the authors have adequately addressed your comments raised in a previous round of review and you feel that this manuscript is now acceptable for publication, you may indicate that here to bypass the “Comments to the Author” section, enter your conflict of interest statement in the “Confidential to Editor” section, and submit your "Accept" recommendation.

Reviewer #1: All comments have been addressed

Reviewer #2: All comments have been addressed

2. Is the manuscript technically sound, and do the data support the conclusions?

Reviewer #1: (No Response)

Reviewer #2: Yes

3. Has the statistical analysis been performed appropriately and rigorously? 

Reviewer #1: (No Response)

Reviewer #2: Yes

4. Have the authors made all data underlying the findings in their manuscript fully available?

Reviewer #1: (No Response)

Reviewer #2: Yes

5. Is the manuscript presented in an intelligible fashion and written in standard English?

Reviewer #1: (No Response)

Reviewer #2: Yes

6. Review Comments to the Author

Reviewer #1: (No Response)

Reviewer #2: (No Response)

7. PLOS authors have the option to publish the peer review history of their article (what does this mean?). If published, this will include your full peer review and any attached files.

Reviewer #1: No

Reviewer #2: No

---

## [Editor Report · Acceptance letter]

24 Mar 2021

PONE-D-20-39915R1 

Inference of stress-strength reliability for two-parameter of exponentiated Gumbel distribution based on lower record values 

Dear Dr. Kılıç Depren:

I'm pleased to inform you that your manuscript has been deemed suitable for publication in PLOS ONE. Congratulations! Your manuscript is now with our production department. 

Kind regards, 

on behalf of

Dr. Feng Chen 

Academic Editor

PLOS ONE